# Thermal Stabilities and Flame Retardancy of Polyamide 66 Prepared by In Situ Loading of Amino-Functionalized Polyphosphazene Microspheres

**DOI:** 10.3390/polym15010218

**Published:** 2022-12-31

**Authors:** Wenyan Lv, Jun Lv, Cunbing Zhu, Ye Zhang, Yongli Cheng, Linghong Zeng, Lu Wang, Changrong Liao

**Affiliations:** 1College of Optoelectronic Engineering, Chongqing University, Shazheng Road 174, Shapingba District, Chongqing 400044, China; 2College of Materials Science and Engineering, Chongqing University of Technology, Hongguang Road 69, Banan District, Chongqing 400054, China

**Keywords:** PA66, polyphosphazene, microsphere, thermal stability, flame retardant

## Abstract

The flame-retardant polyamide 66 composites (FR-PA66) were prepared by in situ loading of amino-functionalized polyphosphazene microspheres (HCNP), which were synthesized in the laboratory and confirmed by a Fourier transform infrared spectrometer (FTIR), scanning electron microscope (SEM), and transmission electron microscope (TEM). The thermal stabilities and flame retardancy of FR-PA66 were measured using thermogravimetric analysis (TGA), a thermogravimetric infrared instrument (TG-IR), the limiting oxygen index (LOI), the horizontal and vertical combustion method (UL-94), and a cone calorimeter. The results illustrate that the volatile matter of FR-PA66 mainly contains carbon dioxide, methane_4_, and water vapor under heating, accompanied by the char residue raising to 14.1 wt% at 600 °C and the value of the LOI and UL-94 rating reaching 30% and V-0, respectively. Moreover, the addition of HCNP decreases the peak of the heat release rate (pHRR), total heat release (THR), mass loss (ML), and total smoke release (TSR) of FR-PA66 to 373.7 kW/m^2^, 106.7 MJ/m^2^, 92.5 wt%, and 944.8 m^2^/m^2^, respectively, verifying a significant improvement in the flame retardancy of PA66.

## 1. Introduction

Polyamide 66 is an important engineering plastic, mainly used in electronic and electrical, transportation, aerospace, and other fields. The LOI of PA66 is 24.0%, with a UL-94 V-2 rating. It is particularly important to improve the flame retardancy of PA66 [1]. Currently, flame retardants are usually added to improve the flame retardancy of PA66, such as halogenated flame retardants decabromodiphenyl ether [2], metal hydroxide [3], nano clay [4], nitrogen compounds [5], and phosphorus-containing organic compounds [6]. Because halogenated flame retardants are prone to produce toxic and corrosive smoke and gas during combustion, they easily corrode equipment, harm the environment, and harm the human body.

To avoid the fire hazards and environmental pollution accompanied by using halogen-based flame retardants in polyamide [7], phosphorus-containing flame retardants have been investigated and found to be effective substitutes with high efficiency, high charring, no melting drip, and economic performance [8]. These characteristics can improve the fire resistance and char formation of polyamide in the combustion process owing to their specific flame retardant mechanism [9]. However, the low degradation temperature of the elementary substance in phosphorus-type flame retardants limits their application in the processing of polyamide [10].

In recent years, polyphosphazene derivatives have been proven to be the most prospective flame retardants owing to their designability and char formation. For instance, novel cross-linked polyphosphazene microspheres were synthesized with 4,4′-dihydroxy biphenyl and tannic acid as co-monomers and decorated with layered double hydroxide to improve the flame retardancy of the epoxy resin. The results revealed that the EP containing 4.0 wt% microspheres exhibited the highest LOI of 29.7 and a UL-94 V-0 rating. Furthermore, the pHRR, THR, and TSR of the epoxy resin composites were significantly reduced and were superior to most of their previously reported counterparts [11]. Moreover, poly-(cyclotriphosphazene-co-4,4-sulfonyldianiline) (PDS) containing amino and hydroxyl groups was synthesized by Z.P Mao and co-workers [12], and the effect of polyphosphazene with different functional groups on the flame retardancy of polyethylene terephthalate was studied. After adding 5 wt% PDS, the LOI of PET composites increased to 33.1%, and they passed the UL-94 V-0 test.

In addition, a series of polyphosphazene flame retardants have been synthesized to improve the flame retardancy, thermal stability, and mechanical properties of polymers by Y Hu and co-workers [13]. For example, a novel allyl-functionalized linear polyphosphazene (PMAP) was designed and synthesized. With the inclusion of 3 wt% PMAP, the pHRR and TSP of composites were reduced by 51.3% and 17.8%, respectively, and the residual char increased significantly as well. Moreover, the impact strength increased by 85.3%, indicating that the toughness was effectively enhanced [14]. Another novel multifunctional organic–inorganic hybrid, melamine-containing polyphosphazene wrapped ammonium polyphosphate (PZMA@APP) with rich amino groups was prepared and used as an efficient flame retardant. The obtained sample passed the UL-94 V-0 rating with a 10.0 wt% addition of PZMA@APP. Notably, the inclusion of incorporating PZMA@APP led to a significant decrease in the fire hazards of EP (75.6% maximum decrease in pHRR and 65.9% maximum reduction in THR) [15].

Because of the excellent natural flame retardant synergies and thermal stability of polyphosphazene, it is believed that an increasingly important role will be played by polyphosphazene modified with nanoparticles in flame retardant applications. For instance, polyphosphazene loading with MoS_2_ nanosheets has been successfully fabricated and significantly improved the flame retardancy of epoxy resin, i.e., 30.7% and 23.6% reductions in pHRR and THR, respectively [16]. Moreover, amino-functionalized carbon nanotubes/polyphosphazene hybrids (AFP@CNTs) were designed and synthesized to enhance the fire safety and mechanical properties of EP. With the addition of 1.5 wt% AFP@CNTs, the pHRR and THR of EP were reduced by 27.6% and 29.0%, respectively. In addition, the impact strength, tensile strength, and storage modulus increased by 65.0%, 29.0%, and 13.2%, respectively [17].

In this study, HCNP was synthesized through polymerization with phosphonitrilic chloride trimer (HCCP) and 4,4′-diaminobenzanilide (DABA). FR-PA66 was prepared by a twin-screw extrusion reaction with the amino groups of HCNP and end carboxyl groups of PA66. The structure of HCNP and the thermal stability, flame retardancy, and toxic properties of FR-PA66 were explored by FTIR, TGA, TG-IR, cone calorimetry, and SEM.

## 2. Results

### 2.1. Characterization of HCNP

#### 2.1.1. FTIR Analysis

The FTIR spectra of HCNP are shown in Figure 1b. The transmittance peaks at 2970 and 2885 cm^−1^ of HCNP are assigned to the -NH_2_ stretching bands of aniline; the peak at 2927 cm^−1^ shows the -NH bonds in the benzamide group; and the NH in-plane bending vibration and the coupling effect between δNH and νC-N of HCNP are at 955 and 1390 cm^−1^, respectively. The peak at 1612 cm^−1^ of HCNP indicates the presence of the C=O bonds of benzamide in the HCNP. The transmittance bands for P=N and P–N stretching vibration in HCNP are at 1197 and 1164 cm^−1^, respectively. The band at 760 cm^−1^ of HCNP coincides with the CH stretching vibration of the para-substitution of a benzene ring.

#### 2.1.2. Microtopography of HCNP

The microtopography of HCNP is shown in Figure 2. The SEM and TEM results clearly showed that the HCNP presents a spherical structure, and the diameter of the HCNP microsphere is about 2 um with a smooth surface. However, also seen are a few irregular edges on the surface of HCNP, as shown in Figure 2a,b, which are inferred to be incomplete reactions of DABA rich in amino groups according to the equation. Moreover, the TEM results (Figure 2d) showed that HCNP microspheres are amorphous.

### 2.2. LOI and UL-94 Test Results

The LOI values and UL94 results with 1.6 mm thickness of FR-PA66 are listed in Table 1, indicating that suitable HCNP contents (9 wt%) induce a higher LOI value of 30%, and UL94 V-0 rating is achieved at the same time.

### 2.3. Thermal Stability of FR-PA66

#### 2.3.1. TGA Analysis

Figure 3a shows the thermal degradation curves of FR-PA66 under a nitrogen atmosphere at a heating rate of 10 °C/min. It is interesting to see that there are two sharp weight loss peaks of FR-PA66 on account of introducing HCNP, shown in Figure 3b.

The initial temperature (defined as 5% mass loss temperature, T_i_) of the pristine PA66 is 387 °C, and the char residue only reaches 2.1 wt% at 600 °C. As for FR-PA66, the T_i_ and first decomposition peak (T_max1_) decrease continuously with the increase in HCNP. When the HCNP content reaches 9 wt%, although the T_i_ and T_max1_ decrease from 387 °C and 432 °C to 360 °C and 400 °C, respectively, the char residues of FR-PA66 increase steadily from 39.65 wt% to 67.5 wt% at T_max1_, and 2.10 wt% to 14.1 wt% at 600 °C.

The data details are noted in Table 2.

#### 2.3.2. TG-IR Analysis

Figure 3c displays the TG-IR curves of FR-PA66 under an inert atmosphere in the range of 300–600 °C. At 380 °C, the band at 669 cm^−1^ was assigned to the aldehyde such as acetaldehyde and aliphatic ketones, and another carbonyl band located at 1508 cm^−1^ was assigned to the amide carbonyl. At 400 °C, the absorbance peaks observed are as follows: 931 cm^−1^ (NO_2_), 966 cm^−1^ (C=N), 1000–1200 cm^−1^ (P=O and C-P=O vibrations), 1350–1700 cm^−1^ (NH and C=O), 1706 and 1769 cm^−1^ (most probably the aliphatic ketone and cyclopentanone, the major degradation product of PA66), 2365 cm^−1^ (CO_2_, confirmed in Figure 3d), and 1454 and 2929 cm^−1^ (the P-alkane) [18,19]. In addition, with gradually increasing temperature, at approximately 470 °C, the peak at 2860 cm^−1^ indicates CH_4_ as the major volatile released (Figure 3d).

### 2.4. The Flame Retardancy of FR-PA66

Cone calorimetry is the most useful technique to evaluate the flame retardancy of materials. The heat release rate (HRR), THR, ML, mass loss rate (MLR), smoke production rate (SPR), and TSR profiles of FR-PA66 are shown in Figure 4. When the content of HCNP reaches 9 wt%, the pHRR declines significantly from 717.5 kW/m^2^ (pristine PA66) to 373.7 kW/m^2^, the THR reduces from 145.1 MJ/m^2^ to 106.7 MJ/m^2^ (Figure 4a), the ML remains at 7.5 wt% when the flame is extinguished, which is 7.0 wt% higher than that of the pristine PA66 (0.5 wt%) (Figure 4b), and the TSR decreases dramatically from 2667.0 m^2^/m^2^ of neat PA66 to 944.8 m^2^/m^2^ of FR-PA66 (Figure 4c).

The data details are noted in Table 3.

### 2.5. Microtopography

Figure 5 shows the SEM pictures of the residual char after the cone calorimetry test of FR-PA66, clearly indicating that the charred layer of FR-PA66 has a denser carbonized state (Figure 5b) than pure PA66 (Figure 5a). Figure 5c,d reveal that the char layer has a dense, high-surface-area, folded structure, and this particular morphology greatly blocks the heat transfer from the combustion of FR-PA66.

## 3. Discussion

### 3.1. Thermal Stability of FR-PA66

Regarding the TGA results of FR-PA66, the T_max1_ gradually shifted to a lower temperature than that of the neat PA66 with increasing HCNP due to the low T_i_ of pure HCNP. Figure 1c shows that the T_i_ of HCNP is 352 °C because of the P-N bond priority breakage between HCCP and DABA under lower temperatures. It is interesting to see that the P-containing alkanes of FR-PA66 are detected by the TG-IR test under 400 °C, which is significantly different from the thermal decomposition products of pure PA66 [20]. Moreover, the R-P· radicals produced by HCNP can catalyze and accelerate the breaking of C-N bonds at the α position of the PA66 molecular chain, resulting in the formation of unstable amino radicals and carbonyl radicals, leading to the decomposition of the FR-PA66. Therefore, the T_i_ decreases significantly in the process of heating because the nitrogen oxide and vapor are easily released accompanying the thermal decomposition of nitrogen-containing compounds at low temperatures [21], which is confirmed by the TG-IR results. Moreover, it is worth noting that FR-PA66 shows two decomposition peaks compared with pure PA66, and the T_max2_ is higher than the T_max1_ of pure PA66 irrespective of the content of HCNP. This phenomenon may be due to the dehydration reaction of CNHO groups and the dehydrogenation reaction of α-CH_2_ facilitated by the R-P· free radical substances produced from HCNP, accelerating the formation of the protective compact char layers [22].

### 3.2. Flame Retardancy of FR-PA66

The flame retardancy of PA66 obviously improved with the introduction of HCNP, and the significant decrease in the HRR, THR, ML, and TSR indicates the enhanced flame retardancy of FR-PA66, which is in keeping with the results of other studies on flame-retardant polyamides [23,24]. This phenomenon is attributed to the abundant release of phosphorous- and carbonyl-containing compounds in the initial heating stage. On the one hand, the R-P· groups (confirmed by the TG-IR results) obtained from HCNP under continuous heating can capture the free radicals produced from the degradation of PA66 and interrupt the chain reaction. On the other hand, the stable multilayer carbonaceous chars (Figure 5) produced from the reaction between HCNP and the decomposition products of PA66 can insulate the underlying substrate from the heat source and slow down both the heat and mass transfer [25]. Thus, it can be summarized that HCNP can effectively improve the flame retardancy of PA66 following the gas-phase flame retardant mechanism.

## 4. Materials and Methods

### 4.1. Materials

HCCP and DABA were provided by energy chemical Co., Ltd. (ShangHai, China). Triethylamine, acetonitrile, and anhydrous ethanol were purchased from Macklin Co., Ltd. (ShangHai, China). PA66 was supplied by Dupont China Co., Ltd. (ShangHai, China).

### 4.2. Synthesis of HCNP

We accurately weighed and dissolved 0.696 g HCCP and 1.362 g DABA in 200 mL acetonitrile; the mixed solution was poured into the three flasks with nitrogen as a protection gas placed in an ultrasonic cleaner under 40 kHz for dispersing at a temperature of 50 °C. Then, 3 mL TEA was added to the three flasks after sonication for 10 min, carrying on the reaction for 4–6 h. The products were washed 3–4 times with anhydrous ethanol and deionized water. Finally, the HCNP was obtained after 24 h at 80 °C in a vacuum drying oven.

### 4.3. Preparation of FR-PA66

PA66 pellets and HCNP powder were dried for 4 h at 100 °C prior to extrusion. The FR-PA66 was prepared by twin-screw extrusion reaction in the range of 260–270 °C. The test samples were manufactured using a TTI-95G injection molding machine under 265 °C with a 180 r/min rotation speed. Figure 6 reveals the microtopography of pure PA66, HCNP, and FR-PA66. It can be seen from Figure 6c that the HCNP microspheres were dispersed more uniformly without obvious phase interfaces, and the peeling phenomenon appeared on the surface of the microspheres (Figure 6d), indicating that the terminal carboxyl groups of PA66 were chemically bonded with the amino groups on the surface of the microspheres during the extrusion reaction, which destroyed the regularity of the surface of the microspheres (Figure 6b, smooth surface and no peeling phenomenon) and led to the core–shell separation of the microspheres (Figure 6d), confirming the occurrence of the in situ loading reaction.

### 4.4. Measurement Methods

FTIR spectra (wavelength range: 4000–500 cm^−1^, resolution: 4.0 cm^−1^) were recorded using a Nicolet iS10 FTIR spectrophotometer by using thin KBr pellets.

TGA was performed using a Q50 thermal gravimetric analyzer (TA instruments, New Castle, DE, America); 3–5 mg samples were heated from 50 to 600 °C at a heating rate of 10 °C/min under a nitrogen atmosphere.

The decomposition products were identified using a Q50 thermogravimetric analyzer coupled to a Nicolet Nexus spectrometer (TG-IR). The sample (approximately 8 mg) was heated in an open alumina crucible from 25 to 600 °C at a heating rate of 10 °C/min.

Cone calorimetry was measured by ENISO1716 (FTT, East Grinstead, England) with a 100 × 100 × 4 mm^3^ sample at 35 kW/m^2^.

The microstructure study of the HCNP and char layers formed during the combustion in the cone calorimetry test was conducted using the scanning electron microscope ZEISS Sigma HDTM. Images were obtained under vacuum at a voltage of 5 kV.

## 5. Conclusions

The FR-PA66 was prepared by twin-screw extrusion reaction in situ loading HCNP. When the content of HCNP reached 9 wt%, the LOI and UL94 values of FR-PA66 improved up to 30% and V-0, respectively. Although the T_i_ and T_max1_ of FR-PA66 decreased by 27 °C and 32 °C, respectively, lower than that of the pristine PA66, the residue at T_max1_ and 600 °C was raised 27.9 wt% and 12.0 wt%, respectively, showing the improvement in the thermal stability of PA66. In addition, the TG-IR analysis showed the formation of nitrogen- and phosphorus-containing radicals under heating, which further confirmed the gas flame retardant mechanism of HCNP. Moreover, the pHRR, THR, and TSP of FR-PA66 decreased 47.9%, 26.5%, and 68.9%, respectively, accompanied by an increase in ML of about 7.0 wt%, revealing that the flame retardancy of PA66 had been modified markedly by adding HCNP. The SEM microtopography images also showed excellent compatibility and dispersion of HCNP in the PA66 matrix.

## Figures and Tables

**Figure 1 polymers-15-00218-f001:**
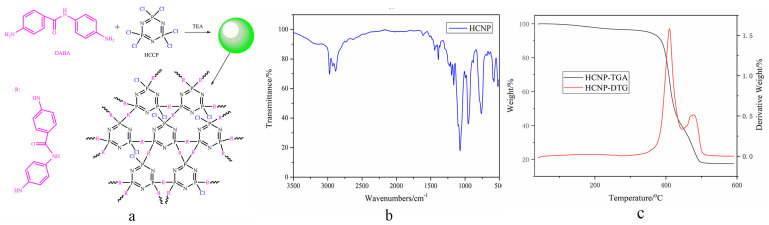
The synthetic routes and FTIR spectra of HCNP. (**a**) The synthetic routes of HCNP, (**b**) the FTIR spectra of HCNP, and (**c**) the TGA curves of HCNP.

**Figure 2 polymers-15-00218-f002:**
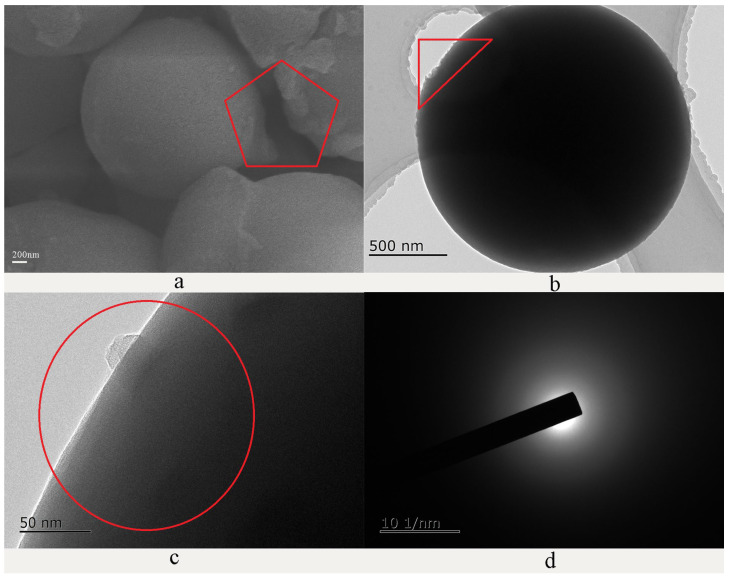
The microtopography of HCNP microsphere. (**a**) The SEM image of HCNP, (**b**–**d**) the TEM images of HCNP.

**Figure 3 polymers-15-00218-f003:**
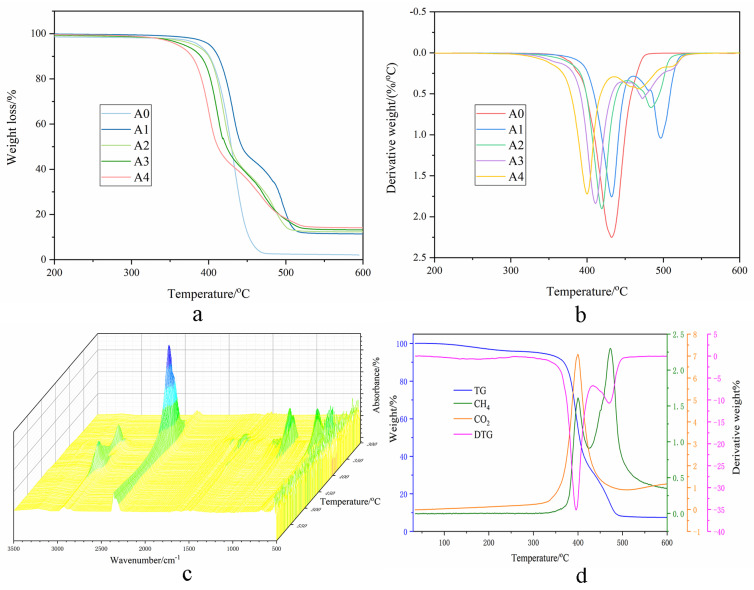
The thermal properties curves of FR–PA66. (**a**) The TGA curves of FR–PA66, (**b**) the DTG curves of FR–PA66, and (**c**,**d**) the TG–IR curves of FR–PA66.

**Figure 4 polymers-15-00218-f004:**
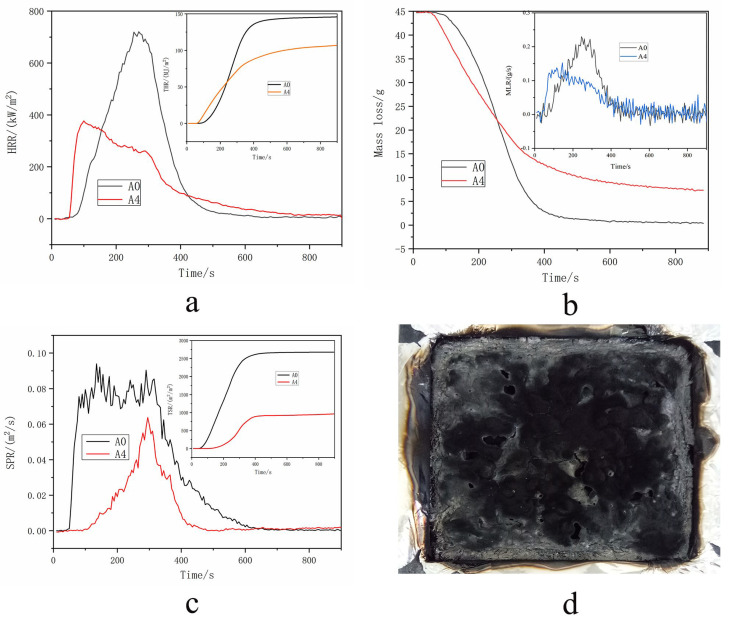
The cone curves of FR–PA66. (**a**) The HRR and THR curves of FR-PA66, (**b**) the MLR and ML curves of FR–PA66, (**c**) the SPR and TSR curves of FR–PA66, and (**d**) a picture of FR–PA66 after the cone test.

**Figure 5 polymers-15-00218-f005:**
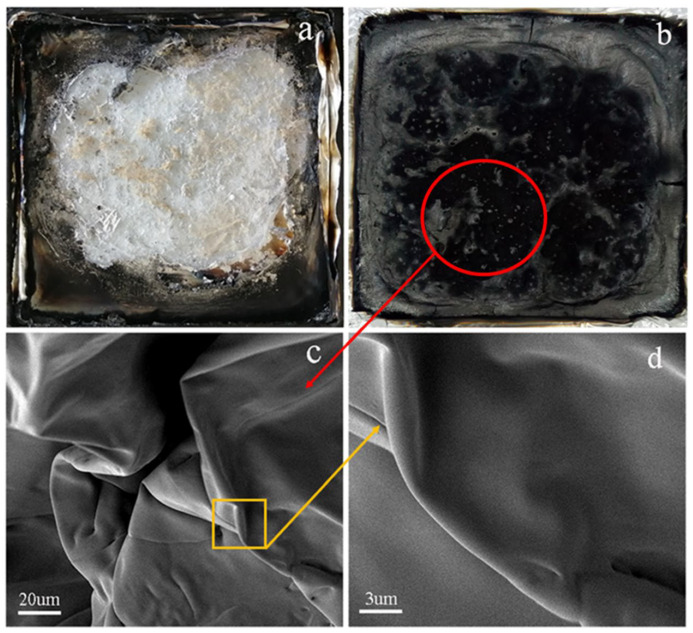
The pictures of residual char and SEM morphology. (**a**) the picture of pure PA66 after Cone calorimetry test; (**b**) the picture of FR–PA66 after Cone calorimetry test; (**c**,**d**) the SEM mophology images of FR–PA66; and red circle and yellow box: Zoom on the specified area.

**Figure 6 polymers-15-00218-f006:**
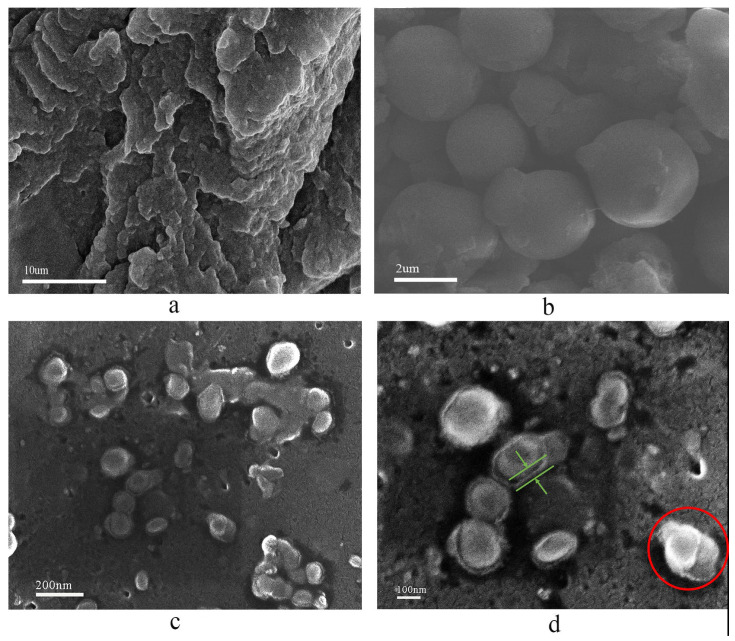
The microtopography of FR–PA66. (**a**) The SEM microtopography of pure PA66, (**b**) the SEM microtopography of HCNP only, and (**c**,**d**) the SEM microtopography of FR–PA66. And green lines and arrows and red circle: the peeling phenomenon on the surface of the microsphere.

**Table 1 polymers-15-00218-t001:** The LOI and UL-94 results of FR-PA66.

Samples	PA66/wt%	HCNP/wt%	LOI	UL94
A0	100	0	24	V-2
A1	97	3	25	V-2
A2	95	5	26	V-2
A3	93	7	28.5	V-1
A4	91	9	30	V-0

**Table 2 polymers-15-00218-t002:** The TGA and DTG detail of FR-PA66.

Samples	T_i_/°C	T_max1_/°C	T_max2_/°C	Residue%/(T_max1_)	Residue% (Final)
A0	387	432	-	39.7	2.1
A1	401	432	497	62.8	11.4
A2	382	419	484	66.0	12.3
A3	368	411	473	67.4	13.2
A4	360	400	467	67.5	14.1

T_max1_ related to the first degradation of FR-PA66. T_max2_ related to the second degradation of FR-PA66.

**Table 3 polymers-15-00218-t003:** The data of cone calorimetry.

Samples	pkHRR/(kW/m^2^)	THR/(MJ/m^2^)	ML/%	TSR/(m^2^/m^2^)
A0	717.5	145.1	99.5	2667.0
A4	373.7	106.7	92.5	944.8

## Data Availability

Data are contained within the article.

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
