# Peer review of "Thermal Stabilities and Flame Retardancy of Polyamide 66 Prepared by In Situ Loading of Amino-Functionalized Polyphosphazene Microspheres"

_polymers, 2022, doi:10.3390/polym15010218_

Round 1

Reviewer 1 Report

In this manuscript, the authors report the thermal stabilities and flame retardant of amino-functionalized polyphosphazene microsphere/polyamide 66 composites. The following comments need to be addressed.

Comment 1. The authors should abbreviate the term when it first appears, e.g. LOI, pHRR

Comment 2. (Figure 6) The SEM images of two control groups (PA66 only and HCNP only, both after the extrusion reaction) should be provided to exclude the possibility that the thermal extrusion process itself deformed the microspheres.

Comment 3. Re-number all the figures, it is weird that Figure 6 appears first.

Comment 4. What A0-A4 stand for should be clearly indicated in section 2.3. The table showing the component of A0-A4 should be before table 1 and Figure 3. Otherwise, readers would have no idea about what A0-A4 are when reading the TGA section.

Comment 5. Table 1. Change T5% to Ti to make all the terms consistent.

Comment 6. What are the y-axes of Figure 3d? How did you quantify the amount of CO2 and CH4 generated?

Comment 7. Change the x-axis of Figure 4: times to time.

Comment 8. The authors should provide instrument and characterization details, they are all missing in this manuscript.

Comment 9. Section 3.3.2 TG-IR analysis. What is the large peak between 2400-2500 cm-1 that increases from 300- 400 oC and then decreases?

Author Response

Thank you for your valuable advice on our paper (ID: polymers-2100240), and we have carefully revised the manuscript after reading reviewers' suggestion.

Comment 1. The authors should abbreviate the term when it first appears, e.g. LOI, pHRR

Reply: All of the abbreviations are abbreviated when it first appears.

Comment 2. (Figure 6) The SEM images of two control groups (PA66 only and HCNP only, both after the extrusion reaction) should be provided to exclude the possibility that the thermal extrusion process itself deformed the microspheres.

Reply: The SEM images of pure PA66 and HCNP only have been added in Figure 6, labeled as Figure 6 a and b, respectively.

Comment 3. Re-number all the figures, it is weird that Figure 6 appears first.

Reply: All the figures are re-numbered in this paper.

Comment 4. What A0-A4 stand for should be clearly indicated in section 2.3. The table showing the component of A0-A4 should be before table 1 and Figure 3. Otherwise, readers would have no idea about what A0-A4 are when reading the TGA section.

Reply: Table 2 in the manuscript has been adjusted before the thermal stability analysis section and renamed Table 1.

Comment 5. Table 1. Change T5% to Ti to make all the terms consistent.

Reply: The T5% in Table 2 (Table 1 in the manuscript) has been changed to Ti.

Comment 6. What are the y-axes of Figure 3d? How did you quantify the amount of CO2 and CH4 generated?

Reply: Units have been added to the Y-axis in Figure 3 d, and the amount of CO2 and CH4 (dimensionless number) is calculated by the Gram-Schmidt reconstruction algorithm based on the recorded infrared spectral data.

Comment 7. Change the x-axis of Figure 4: times to time.

Reply: The x-axes of Figure 4 have been changed to time.

Comment 8. The authors should provide instrument and characterization details, they are all missing in this manuscript.

Reply: The instrument and characterization details have been added in section 4.4 measurement methods.

Comment 9. Section 3.3.2 TG-IR analysis. What is the large peak between 2400-2500 cm-1 that increases from 300- 400 oC and then decreases?

Reply: The large peak between 2400-2500 cm-1 reveals the release of CO2 under heating, which is confirmed by Figure 3 d. And the result has been added to the revised draft.

Reviewer 2 Report

Polyamide 66 and its glass-fibre reinforced composites are widely applied as engineering plastics. Many applications require rendering this polymer and its composites flame-retardant. Due to the high processing temperatures of PA 66 thermal stability of the flame-retardant additives is crucial.

Only a couple of phosphorus-based flame retardants has sufficiently high decomposition temperature to endure the temperature (> 280°C) necessary for processing PA 66. Flame-retardant mixtures based on aluminium diethylphosphinate were found to be suitable for PA 66 composites and are applied in large scale. Nevertheless, there is a demand for improved halogen-free flame-retardant solutions for PA 66.

The presented study deals with investigations on a novel polyphosphazene and its effect on the thermal and flame-retardant properties of PA 66. Fibre-reinforced composites were not considered. In recommend the study for publication after performing revision of the manuscript draft (see below).

First of all, the introduction part should be completed. Whereas sufficient information on the flame-retardant properties of phosphazene derivatives has been given, actual developments of flame retardants for polyamides is missing.  

The polyphosphazene as the subject of investigation was synthesized by a simple procedure using ultrasonics. Investigations of the obtained additive clearly revealed that it has a spherical micro-structure. The novel additive was incorporated into PA 66 using a two-screw lab extruder, whereby relatively low temperature was applied (260-270°C). Specimens for UL 94 test and cone colorimetric investigations were manufactured. Samples without additive were manufactured for comparison. However, the authors did not provide any details what equipment was used and which parameter were applied to prepare the test samples (injection moulding? temperature?). The experimental part should by supplemented accordantly! Investigations by microscopic techniques confirmed evenly distribution of the additive particles inside the polymer matrix.

The thermal behaviour of additive-containing samples was investigated by TGA, DTG and TG-IR. However, the TG-IR results provided in subsection 3.2.2. are not really valuable due to following reasons: detection of methane is not surprising and discussion is missing. Only indication of P-containing alkanes is interesting. The authors should discus the results and compare them with the TG-IR spectra of pure PA66. It is a pity, that the TGA curve of the pure additive is not pictured.

The flame-retardant test results are more interesting: The best rating in the UL94 test (V0) could be achieved at an relatively low additive loading (9 wt.-%). Investigations of cone calorimetry and the morphology of char residue revealed that the additive promotes formation of a dense char residue protecting the underlying material. It is important that the polyphosphazene additive decreases the smoke formation in case of fire.

The authors should improve and supplement the conclusion part because it provides too few information. The headline of the paper appears to me not ideal and should be modified or replaced by a better one.

Last, but not least: the manuscript should be carefully checked again to replace some errors like inappropriate terms, doubled words and gramma errors.

Author Response

Thanks for your positive evaluation on our paper (ID: polymers-2100240), and we have carefully revised the manuscript following the reviewer's suggestion.

Polyamide 66 and its glass-fibre reinforced composites are widely applied as engineering plastics. Many applications require rendering this polymer and its composites flame-retardant. Due to the high processing temperatures of PA 66 thermal stability of the flame-retardant additives is crucial.

Only a couple of phosphorus-based flame retardants has sufficiently high decomposition temperature to endure the temperature (> 280°C) necessary for processing PA 66. Flame-retardant mixtures based on aluminium diethylphosphinate were found to be suitable for PA 66 composites and are applied in large scale. Nevertheless, there is a demand for improved halogen-free flame-retardant solutions for PA 66.

The presented study deals with investigations on a novel polyphosphazene and its effect on the thermal and flame-retardant properties of PA 66. Fibre-reinforced composites were not considered. In recommend the study for publication after performing revision of the manuscript draft (see below).

First of all, the introduction part should be completed. Whereas sufficient information on the flame-retardant properties of phosphazene derivatives has been given, actual developments of flame retardants for polyamides is missing.  

Reply: A brief description of flame-retardant polyamides has been added in the first paragraph of the introduction.

The polyphosphazene as the subject of investigation was synthesized by a simple procedure using ultrasonics. Investigations of the obtained additive clearly revealed that it has a spherical micro-structure. The novel additive was incorporated into PA 66 using a two-screw lab extruder, whereby relatively low temperature was applied (260-270°C). Specimens for UL 94 test and cone colorimetric investigations were manufactured. Samples without additive were manufactured for comparison. However, the authors did not provide any details what equipment was used and which parameter were applied to prepare the test samples (injection moulding? temperature?). The experimental part should by supplemented accordantly! Investigations by microscopic techniques confirmed evenly distribution of the additive particles inside the polymer matrix.

Reply: The equipment and equipment details of test samples have been revised in section 4.3.

The thermal behavior of additive-containing samples was investigated by TGA, DTG, and TG-IR. However, the TG-IR results are provided in subsection 3.2.2. are not valuable due to the following reasons: detection of methane is not surprising and discussion is missing. The only indication of P-containing alkanes is interesting. The authors should discuss the results and compare them with the TG-IR spectra of pure PA66. It is a pity, that the TGA curve of the pure additive is not pictured.

Reply: The discussion of P-containing alkanes and comparison with the pure PA66 have been added in section 3.1, and the TGA curve of HCNP has been pictured in Figure 3 c and analyzed in section 3.1. And the TG-IR of pure PA66

The flame-retardant test results are more interesting: The best rating in the UL94 test (V0) could be achieved at an relatively low additive loading (9 wt.-%). Investigations of cone calorimetry and the morphology of char residue revealed that the additive promotes formation of a dense char residue protecting the underlying material. It is important that the polyphosphazene additive decreases the smoke formation in case of fire.

The authors should improve and supplement the conclusion part because it provides too few information. The headline of the paper appears to me not ideal and should be modified or replaced by a better one.

Reply: The TG-IR results and microtopography details have been provided in the conclusion part following the suggestion of the reviewer.

Last, but not least: the manuscript should be carefully checked again to replace some errors like inappropriate terms, doubled words and gramma errors.

Reply: The English spelling and grammar were revised with the help of the group.

Round 2

Reviewer 1 Report

The authors have addressed all the comments. I suggest this manuscript for publication.

Reviewer 2 Report

The study of Liao et al. has been revised, whereby most of the demanded alterations and supplementations were considered. Most importantly, the abstract part, introduction as well as conclusion were supplemented by essential information. In addition, relevant experimental details were introduced into the manuscript.

Therefore, I recommend the second version of the manuscript for publishing. Please remove a few spelling errors before (e.g.: the headline needs a point after “…Polyamide 66” and “In Situ …”; line 70: please replace “because” by “Because”).